# Genome-Wide Identification of NAC Gene Family Members of Tree Peony (*Paeonia suffruticosa* Andrews) and Their Expression under Heat and Waterlogging Stress

**DOI:** 10.3390/ijms25179312

**Published:** 2024-08-28

**Authors:** Qun Wang, Lin Zhou, Meng Yuan, Fucheng Peng, Xiangtao Zhu, Yan Wang

**Affiliations:** 1Key Laboratory of Tree Breeding and Cultivation of National Forestry and Grassland Administration, Research Institute of Forestry, Chinese Academy of Forestry, Beijing 100091, China; wangqun0024@outlook.com (Q.W.); zhoulin1214@163.com (L.Z.); yuanmeng00829@163.com (M.Y.); pfc2121@163.com (F.P.); 2College of Jiyang, Zhejiang A&F University, Zhuji 311800, China

**Keywords:** tree peony (*Paeonia suffruticosa* Andrews), environmental stress, NAM-ATAF1/2-CUC2 (NAC) gene family, genome-wide identification, expression analysis

## Abstract

An important family of transcription factors (TFs) in plants known as NAC (NAM, ATAF1/2, and CUC2) is crucial for the responses of plants to environmental stressors. In this study, we mined the NAC TF family members of tree peony (*Paeonia suffruticosa* Andrews) from genome-wide data and analyzed their response to heat and waterlogging stresses in conjunction with transcriptome data. Based on tree peony’s genomic information, a total of 48 *PsNAC* genes were discovered. Based on how similar their protein sequences were, these *PsNAC* genes were divided into 14 branches. While the gene structures and conserved protein motifs of the *PsNAC* genes within each branch were largely the same, the cis-acting elements in the promoter region varied significantly. Transcriptome data revealed the presence of five *PsNAC* genes (*PsNAC06*, *PsNAC23*, *PsNAC38*, *PsNAC41*, *PsNAC47*) and one *PsNAC* gene (*PsNAC37*) in response to heat and waterlogging stresses, respectively. qRT-PCR analysis reconfirmed the response of these five *PsNAC* genes to heat stress and one *PsNAC* gene to waterlogging stress. This study lays a foundation for the study of the functions and regulatory mechanisms of NAC TFs in tree peony. Meanwhile, the NAC TFs of tree peony in response to heat and waterlogging stress were excavated, which is of great significance for the selection and breeding of new tree peony varieties with strong heat and waterlogging tolerance.

## 1. Introduction

Environmental stress is a key constraint influencing plant growth, development, and distribution [1,2]. In recent years, climate change has resulted in a surge of extreme weather events, exacerbating damage to plants [3,4]. Tree peony (*Paeonia suffruticosa* Andrews, Paeoniaceae) is a deciduous flowering shrub native to China, now cultivated globally [5,6]. Renowned for its large, vibrant, and aesthetically pleasing blossoms, tree peony is beloved worldwide and is revered as the ‘king of flowers’ in Chinese culture [7,8,9]. Tree peony loves cool conditions, and when the temperature rises beyond 25 °C, their normal growth is halted [10]. In the context of increasing global warming, the growth of tree peony has been greatly affected [11]. Furthermore, tree peony is also intolerant to being waterlogged, but throughout their growth period, particularly during the rainy season, they do get waterlogged frequently, which can seriously hinder their growth [12]. In conclusion, it may be said that heat and waterlogging are two significant environmental conditions that limit tree peony’s growth, which significantly restricts its usage as an attractive plant [13].

Since plants lack the ability to move as animals do, they must adapt to environmental stresses by regulating their gene expression [14]. Genes involved in this response can be classified into two broad categories: functional genes and regulatory genes [15]. Functional genes are directly involved in protecting cells against environmental stresses, such as heat shock proteins [11]. Regulatory genes are involved in regulating the expression of other genes, such as transcription factors (TFs), which recognize cis-acting elements in the promoter region upstream of the 5’ end of the target gene and bind to them directly or indirectly, thereby regulating the expression of the target gene at a specific intensity at a specific time and space [16,17]. NAM-ATAF1/2-CUC2 (NAC) is a plant-specific class of TFs with a conserved DNA-binding structural domain at its N-terminus, consisting of approximately 150 amino acid residues, which can be divided into five sub-structural domains, A–E, and a diverse transcriptional regulatory region at its C-terminus [18,19,20]. Numerous ornamental plants, including the osmanthus [21], plum [22], China rose [23], chrysanthemum [24], and hibiscus [25], have undergone systematic NAC gene family research. However, no comprehensive research has been conducted on NAC TFs in tree peony.

Studies have shown that NAC TFs play important roles in the response to heat and waterlogging stress in a variety of plants. For example, in Arabidopsis, ANAC019 can bind to the promoters of heat responsive genes (such as *HsfA1b* and *HsfB1*) and positively regulate their expression, thereby enhancing heat stress tolerance [26]. In contrast, *ANAC019* mutants exhibit sensitivity to heat stress [27]. Overexpression of *ANAC042* in Arabidopsis enhances tolerance to heat stress, while the opposite phenomenon is observed in knockout plants [28]. The Arabidopsis *ATAF1* mutant and *ANAC055* mutant exhibit enhanced heat stress tolerance, which reveals their negative regulatory roles [29]. Overexpression of maize *ZmNAC074* in Arabidopsis significantly enhances heat stress tolerance [30]. The lily LlNAC014 increases heat stress tolerance by sensing high temperature and translocating to the nucleus to activate the DREB2-HSFA3 module [31]. Rice plants with knock-out and overexpression of *ONAC127* and *ONAC129* both exhibited incomplete grain filling and shriveled grains, which were more severe under heat stress [32]. An Arabidopsis ANAC013 TF can bind to the promoters of multiple hypoxia core genes and positively regulate their expression, thereby enhancing tolerance to waterlogging stress. Knocking out *ANAC013* results in reduced tolerance to waterlogging stress [33]. It is thus conceivable that members of the NAC TF family function in tree peony in response to heat and waterlogging stress.

The mechanism of NAC TFs on tree peony in response to heat stress and waterlogging stress is still unknown, as there are no studies addressing this aspect. Currently, the whole genome sequence of tree peony has been assembled, which makes it possible to systematically analyze NAC TF family members in tree peony and explore NAC TFs associated with heat and waterlogging stress responses [6]. For ornamental plants planted in the ground, it is the traits of the aboveground parts that are the focus of these studies. Therefore, this present study was conducted on leaves. The transcriptomes of tree peony leaves subjected to heat and waterlogging stress were analyzed to explore the key genes involved in the response of tree peony to heat and waterlogging stresses. The primary goal of this study was to lay the foundation for further research on the functions and regulatory mechanisms of tree peony NAC TFs from the perspective of the NAC TF family. Meanwhile, the key NAC TFs identified in tree peony’s response to heat stress and waterlogging stress are of great significance for the selection and breeding of new ornamental tree peony varieties with strong heat and waterlogging tolerance. 

## 2. Results

### 2.1. Identification of PsNAC Family Members and Analysis of Physicochemical Properties

Through the identification of both Blast and Pfam, as well as the tests of CDD, SMART, and HMMER, 48 tree peony NAC genes were finally obtained from the tree peony whole-genome data, which were named as *PsNAC01*-*PsNAC48* (Appendix A). The number of amino acids encoded by the 48 *PsNAC* genes ranged from 135 to 597, with an average of 319.54. The molecular weights ranged from 15,344.43 kDa to 67,955.26 kDa, with an average of 36,368.52 kDa. Isoelectric points ranged from 4.47 to 9.71, with an average of 7.16. The aliphatic amino acid indices ranged from 53.95 to 81.12, with an average of 66.05. The instability indexes ranged from 1.5 to 2.0. The average was 66.05. The instability index indicated that seven proteins were stable (instability index < 40) and the rest were unstable. All proteins were hydrophilic since their average hydrophilicity value was negative. The protein tertiary structure revealed that these proteins had a similar overall structure, but that there were variances in the specific structural elements, which could account for the functional diversity of these proteins (Appendix A). The 48 proteins are primarily found in the nucleus, with a small number also present in the cytoplasm, mitochondria, and other locations, according to subcellular localization predictions (Appendix A).

### 2.2. Phylogenetic Analysis and Classification of PsNAC

To clarify the taxonomic and evolutionary relationships of tree peony NAC gene family members, a neighbor-joining phylogenetic tree was constructed by combining 48 proteins with the NAC proteins from Arabidopsis. According to Ooka, et al. [18], the NAC of Arabidopsis are classified into 16 subgroups. These NAC proteins are divided into 6 groups with 15 branches, based on the phylogenetic analysis (Figure 1): 1A–1B, 2A–2C, 3A–3C, 4A–4D, 5A, and 6A–6B. The OsNAC7, NAP, NAC2, TERN, ANAC001, ANAC063, and ONAC003 subgroups of Arabidopsis are represented by branches 1A, 2A, 3A, 3B, 4A, 4B, and 4C, respectively. Branches 1B, 2B, 2C, and 3C each correspond to two subgroups of Arabidopsis, NAM and NAC1, ATAF and AtNAC3, SENU5 and ONAC022, and TIP and OsNAC8. Branches 4D, 6A, and 6B do not correspond to members of the subgroups in Arabidopsis. 

### 2.3. Protein Conservation Motif and Gene Structure Analysis of PsNAC Family

To understand the structural diversity of these proteins, 20 conserved motifs of 48 proteins were successfully predicted, named Motif 1–Motif 20 (Figure 2d). All 48 protein sequences featured Motif 1, and the majority also contained Motifs 2, 3, 4, 5, 6, 7, and 10 (Figure 2b). Exon and intron length and positions were largely conserved within the same branch of the gene structure, although they varied between branches (Figure 2c). Similar gene structures and conserved motif composition among *PsNAC* genes on the same branch of the evolutionary tree significantly suggest the accuracy of the phylogenetic analysis results (Figure 2a). 

### 2.4. Analysis of Cis-Acting Elements in the Promoter Region of PsNAC Genes

In order to figure out the potential functions and regulatory mechanisms of these proteins, 22 kinds, totaling 723 cis-acting elements in the 2000 bp region upstream of the promoters of all *PsNAC* genes, were successfully predicted (Figure 3). These cis-acting components can be divided into the following five categories: those that are responsive to stress, hormones, development, binding sites, and MYB-related. All *PsNAC* genes had elements that responded to stress (light, low temperature, trauma, defense stress, and hypoxia induction), 47 had elements that responded to hormones (abscisic acid, gibberellin, auxin, salicylic acid, and methyl jasmonate), and 35 had elements that responded to development (cell cycle regulation, endosperm expression, meristem expression, seed-specific, differentiation of the palisade mesophyll cells, and root-specific). In addition, MYB-related response elements (drought induction, flavonoid synthesis, and light response) are present in 34 *PsNAC* genes. 

### 2.5. Expression Pattern Analysis of PsNAC Genes

To clarify the potential roles of the *PsNAC* gene, the expression pattern of *PsNAC* genes in response to heat and waterlogging stress were investigated utilizing existing transcriptomic data. The majority of *PsNAC* genes were found to be expressed in the leaves of the tree peony, while a minority were not expressed, which is in line with the theory of selective expression of genes; after all, not all genes are expressed in every plant tissue (Figure 4). When screening for differentially expressed genes with the thresholds of fold change ≥ 2 or fold change ≤ 0.5 and q-value ≤ 0.01, one *PsNAC* gene (*PsNAC06*) was found to be upregulated and four *PsNAC* genes (*PsNAC23*, *PsNAC38*, *PsNAC41*, *PsNAC47*) were found to be downregulated under heat stress (Table 1). Under waterlogging stress, one *PsNAC* gene (*PsNAC37*) was identified as upregulated, and no downregulated *PsNAC* genes were detected (Table 1).

### 2.6. Annotation and Enrichment Analysis of PsNAC Genes

To deeply explore the functions of *PsNAC* genes, 30 out of 48 *PsNAC* genes were successfully annotated into 20 terms in molecular functions (MF), cellular components (CC) and biological processes (BP) (Figure 5). Among these, 6 *PsNAC* genes with differential expression in response to heat or waterlogging stress were enriched in 15 terms. Five *PsNAC* genes with differential expression in response to heat or waterlogging stress were enriched in terms related to DNA-binding TF activity (GO:0003700) and transcription regulator activity (GO:0140110) in MF, which is consistent with the TFs and their functional descriptions. Two *PsNAC* genes with differential expression in response to heat or waterlogging stress were solely enriched in terms related to the nucleus (GO:0005634) in CC, which is in line with the fact that eukaryotic TFs frequently bind DNA in the nucleus to control gene transcription. In BP, *PsNAC* genes with differential expression in response to heat or waterlogging stress were mostly enriched in terms related to nucleobase-containing compound metabolic process (GO:0006139), biosynthetic process (GO:0009058), and metabolic process (GO:0008152). Additionally, one *PsNAC* gene with differential expression in response to heat or waterlogging stress was also enriched to a certain extent in the term ‘response to abiotic stimulus’ (GO:0009628), which reflects the possibility of its involvement in stress response. However, there are still some *PsNAC* genes that are not annotated to GO terms, and in addition, none of the 48 *PsNAC* genes had any KEGG pathway annotations, according to the results of KEGG annotation. It may not be precise and efficient enough, or the database may not be complete enough; after all, identical annotations in other studies have frequently produced no or few findings.

### 2.7. qRT-PCR Validation of Differentially Expressed PsNAC Genes

To further validate the responsiveness of the six *PsNAC* genes to heat and waterlogging stress, their expression patterns were re-examined using qRT-PCR. The results revealed that the *PsNAC06* gene was markedly upregulated by heat stress, whereas heat stress significantly downregulated the expression of *PsNAC23*, *PsNAC38*, *PsNAC41*, and *PsNAC47* genes. Additionally, waterlogging stress notably upregulated the PsNAC37 gene (Figure 6). These results are consistent with the transcriptome data, confirming the accuracy of the transcriptome data and indicating that these six *PsNAC* genes may play a role in the response of tree peony to heat stress or waterlogging stress. 

## 3. Discussion

The NAC family of structural domain proteins is a plant-specific family of TF proteins, and the first NAC TF family member, NAM, was first identified in Petunia by Souer, et al. [34]. Over the next two decades, researchers have identified NAC family members from a variety of plants, such as 117 in Arabidopsis [35], 85 from asparagus [36], 80 from buckwheat [37], 68 from *Kandelia obovata* [38], 57 from *Simmondsia chinensis* [39], 50 from ginkgo [40], and 37 from woodland strawberry [41]. In this study, we identified 48 *PsNAC* genes in tree peony, which is a lower count compared to most other known plants. This discrepancy is rationalized by the possibility that more replication events may have occurred in other species since their divergence from the common ancestor [25,37]. Additionally, due to substantial duplications or errors, many of the initially identified TFs in plants have been progressively winnowed out in subsequent research. For example, while 117 NAC TFs were initially identified in Arabidopsis, only 100 have been retained to date. The poorly assembled genome of tree peony resulted in an incomplete genome map, preventing the identification of the chromosomal distribution of *PsNAC* genes [6]. Therefore, while removing duplicate genes, multiple copy genes were also removed, resulting in a decrease in the number of *PsNAC* genes identified in tree peony. In the future, if a high-quality chromosome-level genome of tree peony becomes available, the chromosomal distribution of *PsNAC* genes should be re-examined to accurately determine the number of *PsNAC* genes. 

Members of a subgroup that are clustered together may play related or similar roles based on their evolutionary ties. For instance, members of the ATAF, NAP, and AtNAC3 subfamilies of the *Arabidopsis thaliana* NAC TF family play crucial roles in stress responses [2]. There are studies that have successfully identified key genes through phylogenetic relationships [42]. However, there are also studies that have found that genes with similar functions do not cluster together [9]. This may be the result of gene evolution [43]. Although members of a gene family may have similar functions, over time, these functions may differentiate, allowing certain members to perform unique roles under specific environmental conditions, leading to genes with similar functions not clustering together. Moreover, plant stress responses involve numerous distinct pathways [44]. Predicting plant stress response genes solely based on evolutionary relationships may not be a reliable method. Similarly, although GO analysis results show that only one differentially expressed *PsNAC* gene is enriched in the term ‘response to abiotic stimulus’, this does not imply that other differentially expressed *PsNAC* genes do not play a role in the response of tree peony to heat or waterlogging stress. Stress response involves various different biological processes, including signal transduction, transcriptional regulation, and metabolic changes. These processes may be scattered across different pathways in the GO database, rather than concentrated in a specific stress-related pathway. According to studies [45,46], variations in promoter regions are frequently linked to variations in gene activity. Cis-elements in promoter regions play a significant role in controlling gene expression during growth, development, and environmental changes [47]. All *PsNAC* gene promoters in this study contained stress-responsive elements, and the majority of *PsNAC* genes also contained hormone-responsive elements and development-related response elements, which is consistent with earlier findings that NAC TFs are intimately connected to plant growth and development, signal transduction, and stress response [19]. 

According to Nuruzzaman, et al. [48], numerous NAC TFs have been identified as enhancing plant adaptability to environmental challenges and playing a crucial role in shaping plant responses to environmental stimuli. Theoretically, NAC TFs involved in resisting environmental stress should be classified into two categories: up-regulated and positively regulated, or down-regulated and negatively regulated. The former are induced during environmental stress and activate downstream functional genes. For example, maize ZmNAC074 TF is induced by high temperature stress and enhances the heat tolerance of plants under high temperature stress by activating the expression of ROS scavenging genes, heat shock response-related genes, and unfolded protein response-related genes [30]. Waterlogging stress induces the expression of the Arabidopsis ANAC013 TF, which enhances tolerance to waterlogging stress by activating the expression of multiple hypoxia core genes [33]. In this study, *PsNAC06* was activated after heat stress, and *PsNAC37* was activated after waterlogging stress. Therefore, it can be reasonably inferred that *PsNAC06* and *PsNAC37* play positive regulatory roles in the resistance of tree peony to heat and waterlogging stresses, respectively. The latter should be repressed in response to environmental stress, thus relieving its inhibition on downstream functional genes. In this study, *PsNAC23*, *PsNAC38*, *PsNAC41*, and *PsNAC47* were suppressed after heat stress, indicating that they may play a negative regulatory role in the resistance of tree peony to heat stress. However, this is not always the case, as they can be activated in the short term and then gradually repressed as the stress continues. For example, the apple stress resistance gene *MdDREB2A* is negatively regulated by the MdNA029 TF, which is activated under short-term stress, and its expression gradually decreases as the stress persists [49]. This may be because the same TF participates in the regulation of multiple seemingly different processes as a positive or negative regulator [48,50,51]. Short-term stress results in the suppression of some metabolic activities, which are then restored by plants by activating these TFs. However, as long as the stress persists, plants must also tend to reduce the metabolic processes in order to withstand the stress. In summary, the above genes appear to have played a role in the response of tree peony to heat stress or waterlogging stress, but their specific functions still need further research to verify.

## 4. Materials and Methods

### 4.1. Identification and Testing of Transcription Factors

The Arabidopsis NAC gene family protein sequences from the TAIR database (https://www.arabidopsis.org/ (accessed on 1 August 2023) were used as reference sequences for Blast scanning of tree peony genomic data [52]. The Hidden Markov Model (PF 02365) from the Pfam database (http://pfam.xfam.org/ (accessed on 1 August 2023) was used as the query sequence to screen the tree peony genomic data [53]. Both the Blast and Pfam screenings employed a threshold of E-value ≤ 1 × 10^−10^. To check for the presence of NAM structural domains, the identification findings from Blast and Pfam were merged and then examined using CDD (https://www.ncbi.nlm.nih.gov/ (accessed on 1 August 2023) [54], SMART (http://smart.embl-heidelberg.de/ (accessed on 1 August 2023) [55], and HMMER (https://www.ebi.ac.uk/ (accessed on 1 August 2023) [56]. The whole-genome data for tree peony were obtained from the National Gene Bank of China (https://db.cngb.org/ (accessed on 1 August 2023) [6].

### 4.2. Analysis of Physicochemical Properties of Protein Sequences

The basic physicochemical properties of the tree peony NAC protein sequence, including amino acid number, molecular weight, isoelectric point, instability index, aliphatic index, and grand average of hydropathicity, were predicted using the online tool ProtParam (https://www.expasy.org/ (accessed on 10 August 2023) [57]. Using the online tool SWISS-MODEL (https://swissmodel.expasy.org/ (accessed on 16 August 2023), protein tertiary structure was calculated [58]. For the purpose of predicting subcellular localization, the website CELLO (http://cello.life.nctu.edu.tw/ (accessed on 27 August 2023) was employed [59]. 

### 4.3. Construction of Phylogenetic Tree

Phylogenetic trees were constructed using the MEGA 7.0 software [60]. Musle was used to perform sequence alignments [61]. Neighbor-joining (NJ) phylogenetic trees were created using 1000 bootstrap replicates [62]. To visualize and landscape the phylogenetic trees, the online tool iTOL (https://itol.embl.de/ (accessed on 6 September 2023) was employed [63].

### 4.4. Conserved Sequence Comparison and Motif Analysis

The MEME website (https://meme-suite.org/ (accessed on 6 September 2023) was used to search for conserved motifs, with the maximum number of motifs set to 20 and the other parameters left as default [64]. The tree peony NAC gene structure is described in the genome GFF3 file, which may be used to locate introns and exons. TBtools v1.128 was used to visualize conserved motifs and gene architectures [65]. 

### 4.5. Analysis of Cis-Acting Elements in the Promoter Region

The prediction of cis-acting elements in the promoter region of each *PsNAC* gene was conducted using the online program PlantCARE (http://bioinformatics.psb.ugent.be/webtools/plantcare/html/ (accessed on 7 September 2023) [66]. TBtools v1.128 was used to extract the 2000 bp sequence upstream of the start codon of each *PsNAC* gene and to visualize the condensed cis-acting elements. 

### 4.6. Gene Expression Analysis

The National Center for Biotechnology Information (NCBI) (https://www.ncbi.nlm.nih.gov/ (accessed on 20 July 2023) provided the sequence reads archive (SRA) of transcriptome data of tree peony under heat and waterlogging stress. Gene expression levels were described using TPM (transcripts per million) [67]. Heat mapping of gene expression and upstream and downstream analysis of the transcriptome data were conducted using TBtools v1.128.

### 4.7. Annotation and Enrichment Analysis in GO and KEGG Databases

All *PsNAC* genes were GO and KEGG annotated using the online resource EGGNOG (http://eggnog-mapper.embl.de/ (accessed on 10 September 2023) [68]. For the analysis of enrichment and to simplify the annotation files, TBtools v1.128 was employed. The findings of the enrichment analysis were visually presented using the online plotting program ChiPlot (https://www.chiplot.online/ (accessed on 10 September 2023).

### 4.8. Plant Material and Stress Treatments

Twelve tree peony cultivars of *P. suffruticosa* ‘Luoyanghong’ (procured from Heze Guyu Peony Biotechnology Co., Ltd., Heze, China), were cultivated in pots of 21 cm in diameter. For five days, seedlings were initially domesticated under standard field water and fertilizer management in a greenhouse (18–25 °C). Both heat and waterlogging stress treatments were carried out according to the previous research experience [11,13]. The six pots of domesticated seedlings were subjected to heat stress treatment by dividing them into experimental and control groups. The experimental and control groups were cultured in a greenhouse at 27 °C–35 °C and 18 °C–25 °C, respectively, under standard field water and fertilizer management. After three days, the upper healthy leaves were collected, quick-frozen in liquid nitrogen, and stored at −80 °C in a refrigerator for subsequent research. In the waterlogging stress treatment, six pots of domesticated seedlings were divided into experimental and control groups. The control group was managed in accordance with standard field water and fertilizer management practices. The experimental group was flooded to a depth just below the soil surface. After three days, the upper healthy leaves were promptly collected, quick-frozen in liquid nitrogen, and stored in a −80 °C freezer for subsequent research. 

### 4.9. qRT-PCR Analysis

According to the instructions, total leaf RNA was extracted using the Quick RNA Isolation Kit (Huayueyang Biotechnology Co., Ltd., Beijing, China). The EasyScript One-Step DNA Removal and cDNA Synthesis SuperMix Kit (Beijing TransGen Biotech Co., Ltd., Beijing, China) was used to reverse-transcribe the RNA from each sample. The TB Green^®^ Premix Ex Taq^™^ Kit (Beijing TaKaRa Biomedical Technology Co., Ltd., Beijing, China) was used to conduct the qRT-PCR test utilizing a LightCycler^®^ 480 II real-time fluorescence quantitative PCR machine (Roche, Basel, Switzerland). Gene-specific primers were designed with Primer3Plus (https://www.primer3plus.com/ (accessed on 16 September 2023) [69], and the results are provided in Appendix A. The tree peony *Ubiquitin* gene was used as an internal reference gene. Each gene underwent three replications, and the 2^−∆∆CT^ method was used to calculate the relative expression [70]. IBM SPSS Statistics 21.0 software (IBM Corporation, Armonk, NY, USA) was used for the analysis of the relative expression. OriginPro 2021 software (OriginLab Co., Northampton, MA, USA) was employed for the visualization of the relative expression results.

## 5. Conclusions

The physicochemical characteristics, sequence homology, gene structure, promoter elements, functional annotation, and expression patterns of tree peony NAC gene family members were examined in depth in this work using the entire genome data of the plant. According to the phylogenetic tree, a total of 48 tree peony *PsNAC* genes were discovered and divided into 6 groups and 14 branches. Based on the transcriptome and qRT-PCR analyses of *PsNAC* genes under heat and waterlogging stress, five *PsNAC* genes may be involved in the response of tree peony to heat stress and one *PsNAC* gene may be involved in the response of tree peony to waterlogging stress. These results contribute to further exploration of the response mechanism of tree peony to heat and waterlogging stresses.

## Figures and Tables

**Figure 1 ijms-25-09312-f001:**
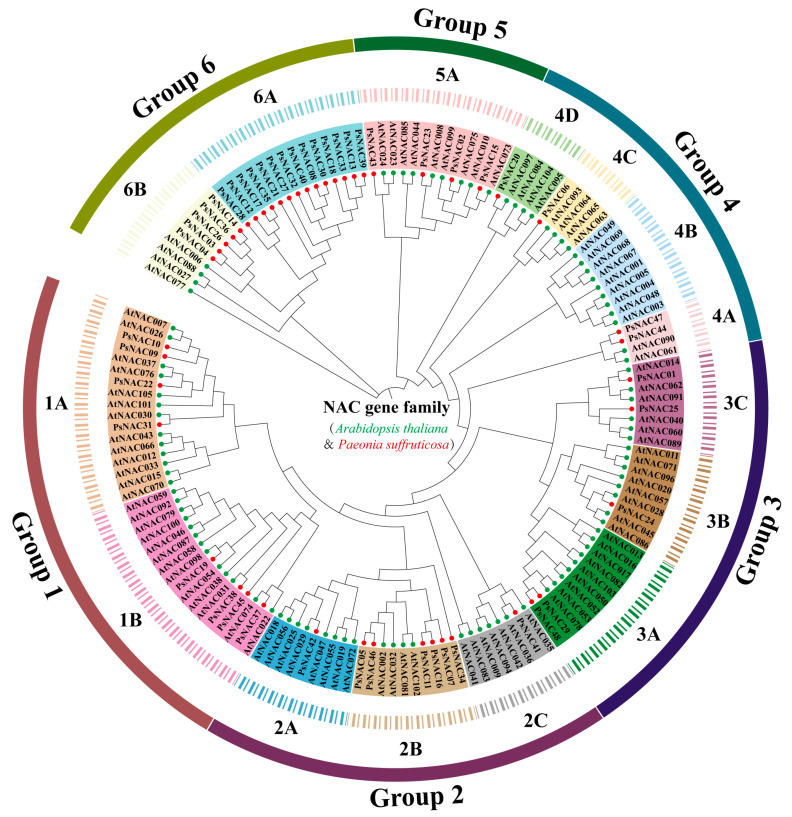
Phylogenetic tree of NAC proteins from tree peony, Arabidopsis, and other species. Each branch is distinguished by a different color and is labeled. Red circles indicate tree peony and green circles indicate Arabidopsis.

**Figure 2 ijms-25-09312-f002:**
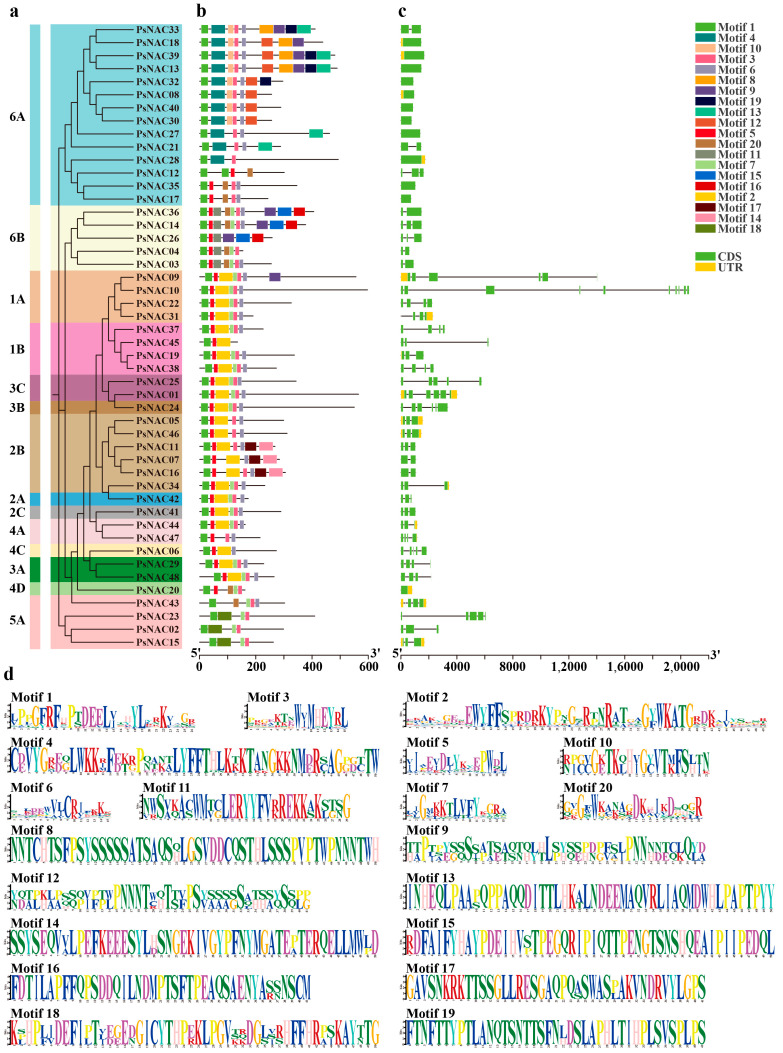
Phylogenetic tree, conserved protein motifs and gene structural features of tree peony NAC genes. Phylogenetic tree (**a**), conserved protein motifs (**b**), gene structure (**c**), sequence logo of the proteins motifs (**d**). The height of each amino acid represents the relative frequency of the amino acid at that position.

**Figure 3 ijms-25-09312-f003:**
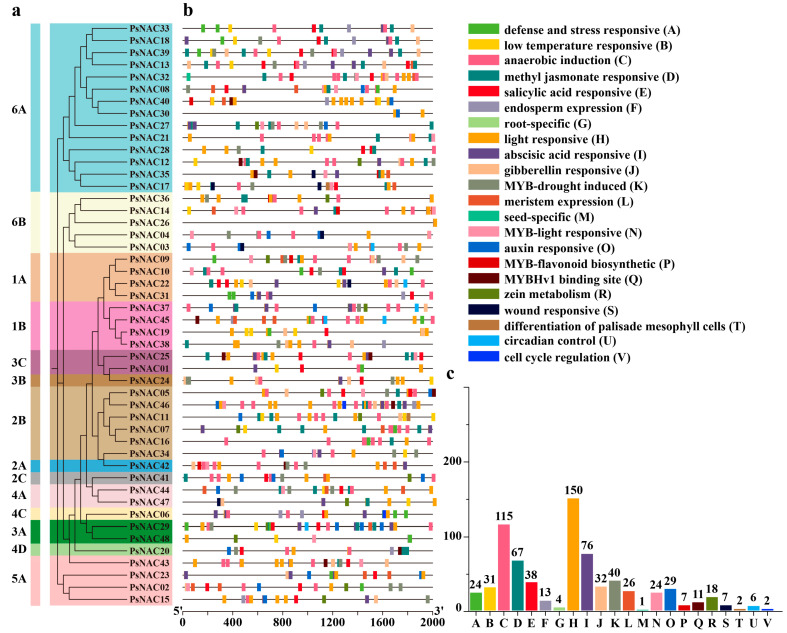
Predicted cis-acting elements of the promoter sequence of the tree peony NAC gene. Phylogenetic tree (**a**), cis-acting element distribution (**b**), number of cis-acting element genes (**c**).

**Figure 4 ijms-25-09312-f004:**
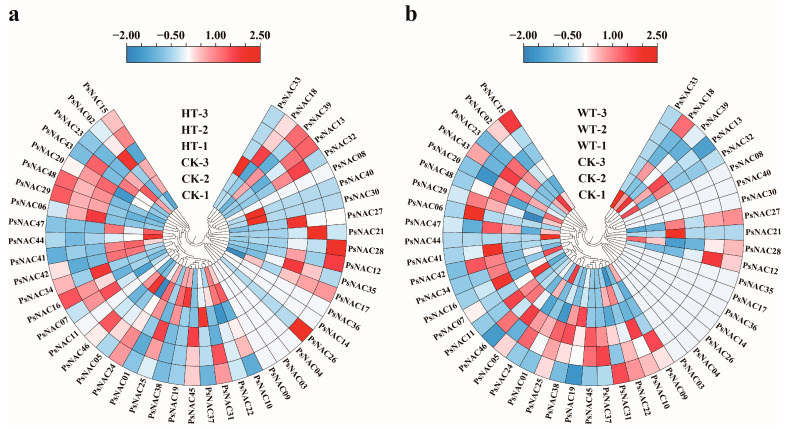
Expression profiles of *PsNAC* genes under heat stress and waterlogging stress in tree peonies. Expression profiles under heat stress (**a**); Expression profiles under waterlogging stress (**b**). CK, control group; HT, heat stress experimental group; WT, waterlogging stress experimental group.

**Figure 5 ijms-25-09312-f005:**
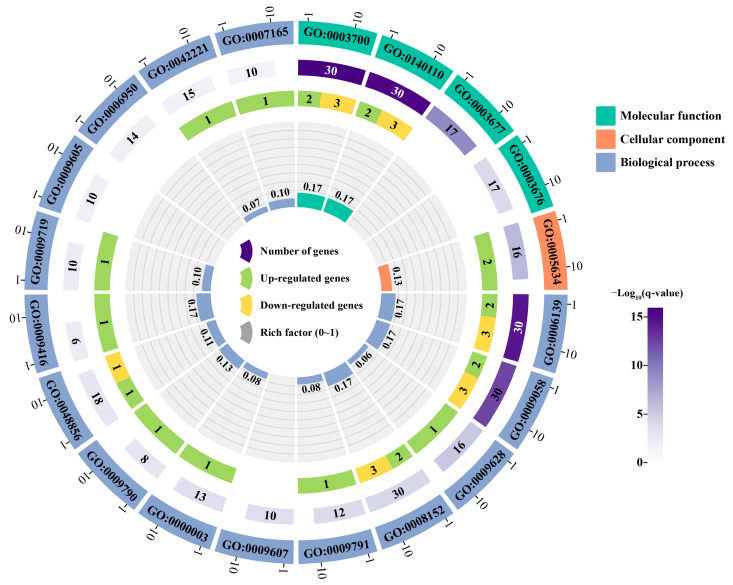
*PsNAC* genes enriched in GO terms. The three circles from the outside in indicate the ID of the GO term, the number of background *PsNAC* genes enriched in that term in the genome, and the number of *PsNAC* genes that respond to heat or waterlogging stress enriched in that term, respectively.

**Figure 6 ijms-25-09312-f006:**
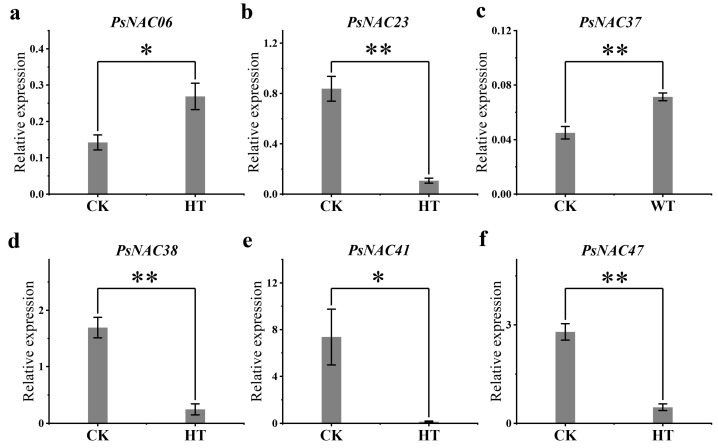
qRT-PCR analysis of the expression patterns of six *PsNAC* genes in response to heat or waterlogging stress treatments. Expression pattern of *PsNAC06* gene under heat stress (**a**), expression pattern of *PsNAC23* gene under heat stress (**b**), expression pattern of *PsNAC37* gene under waterlogging stress (**c**), expression pattern of *PsNAC38* gene under heat stress (**d**), expression pattern of *PsNAC41* gene under heat stress (**e**), expression pattern of *PsNAC47* gene under heat stress (**f**). CK, control group; HT, heat stress experimental group; WT, waterlogging stress experimental group. The values shown are the means ± SE. The *t*-test significance is indicated by an asterisk: * means *p* < 0.05, ** means *p* < 0.01.

**Table 1 ijms-25-09312-t001:** Differential expression of *PsNAC* genes in tree peonies under heat stress and waterlogging stress.

Treatment	Gene Name	Expression Level	log_2_ (Fold Change)	q-Value
Heat stress	*PsNAC06*	up	3.80155351	0.000157825
	*PsNAC23*	down	−1.142697267	3.12 × 10^−9^
	*PsNAC38*	down	−6.598682031	0.0000937
	*PsNAC41*	down	−2.098132113	5.41 × 10^−9^
	*PsNAC47*	down	−2.611193255	0.0000686
Waterlogging stress	*PsNAC37*	up	4.289772524	3.41 × 10^−5^

Note: ‘up’ indicates up-regulation of gene expression compared to control group, and ‘down’ indicates down-regulation of gene expression compared to control group.

## Data Availability

All data from this study are available. The National Gene Bank of China (CNGB) has tree peony whole-genome information that may be accessed by using the accession number CNP0000281. The National Center for Biotechnology Information (NCBI) has transcriptome data accessible with accession number PRJNA802352 and PRJNA522665. The Appendix A and this article provide the remaining data.

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
