# Peer review of "Genome-Wide Identification of NAC Gene Family Members of Tree Peony (Paeonia suffruticosa Andrews) and Their Expression under Heat and Waterlogging Stress"

_ijms, 2024, doi:10.3390/ijms25179312_

Round 1
Reviewer 1 Report
Comments and Suggestions for Authors
NAC is a kind of transcriptional factor involved in various growth process and also plays crucial roles in response to different stresses. The authors identified 48 NACs from the whole genome of Tree Peony according to the genome data. The response of the members to heat stress and waterlogging was also investigated through RNA-seq data. The topic is suited for this journal and the study shows potential, I do have some queries I'd like the authors to improve, and explain or edit.
Major concerns:
1. The analysis was too simple and should be further improved, otherwise, it’s hard to be accepted by the journal.
2. To attract more readers, the writing should be expanded with common issues, or plant species, especially in the introduction and discussion part.
3. There were 5 subgroups of NACs according to the previously investigations, but 6 classes were found here with much less members, how to explain the results?
4. The language should be well polished, it’s hardly to follow.
5. The repetitiveness of the RNA-seq data is too less to be convinced according to the Fig. 4.
Minor concerns:
More detailed information is suggested to be added into the section of “Materials and Method”.
Comments on the Quality of English LanguageThe language should be well polished.
Reviewer 2 Report
Comments and Suggestions for Authors
The manuscript titled "Genome-Wide Identification of NAC Gene Family Members of Tree Peony (Paeonia suffruticosa) and Their Expression Under Heat and Waterlogging Stress" presents a comprehensive analysis of the NAC (NAM, ATAF1/2, and CUC2) gene family in tree peony, with a particular focus on their expression profiles under abiotic stress conditions such as heat and waterlogging. The authors conducted a genome-wide identification of NAC genes, characterized their structural features, and evaluated their expression patterns under stress conditions to elucidate their potential roles in stress responses.
General comments:
The MS provides a comprehensive genome-wide identification and analysis of NAC genes in tree peony, employing a well-executed approach to elucidate the roles of NAC gene families in stress responses. The MS describes Insilco analysis and gene validation of the NAC. The MS well crafted, study gap properly described, material & methods clearly described and well written, results section comprehensively describes, and discussion section also clearly point the problems and discussed. The manuscript would be significantly strengthened and could make a notable impact in the field of plant genomics and stress biology. Below are some points need to respond before publishing to IJMS.
Specific comments:
Line 13: used full form at first site and then abbreviation such as Transcription factor and (TF) rest of the MS.
141: MYB- -related removed (-)
Line 155-158 sentence need revision for grammatically error.

Reviewer 3 Report
Comments and Suggestions for Authors
A review of the manuscript by "Genome-Wide Identification of NAC Gene Family Members of Tree Peony (Paeonia suffruticosa) and Their Expression Under Heat and Waterlogging Stress".
I think that the submitted manuscript should be treated as a communication and not as an original article. It's a short work. The topic is interesting but the authors treated it briefly.
Notes:
The authors write that they used "The National Gene Bank of China (CNGB) (https://db.cngb.org/) provided the tree peony whole genome data"
Was the genome information about a wild species or a cultivar or variety?
If wild species information was used then why did the authors use a variety in the experiments?
After all, the genome of a wild species can differ significantly from a cultivated (modified) variety.
I believe that the experiments should be repeated using a botanical species.
The authors have a lack of education (or laziness) when it comes to using Latin names. Please correct all errors. Why the authors' species names are missing next to the names?
In their conclusion, the authors write: "These results contribute to our understanding of the response mechanisms of tree peony to heat and waterlogging stresses." But are they sure? this research is just a contribution.
Why not do a paper in which the study of gene expression would be combined with the study of physiology and anatomy , such a paper would really reflect what happens to the plant during stress.
Round 2
Reviewer 1 Report
Comments and Suggestions for Authors
I didn't think the authors well addressed the concerns, especially about the comments of 2, 3, and 6.
2. The analysis was too simple and should be further improved, otherwise, it’s hard to be accepted by the journal.
3. To attract more readers, the writing should be expanded with common issues, or plant species, especially in the introduction and discussion part.
6. The repetitiveness of the RNA-seq data is too less to be convinced according to the Fig. 4.
Comments on the Quality of English LanguageIt's improved by still not native.
Reviewer 3 Report
Comments and Suggestions for Authors
The authors made corrections that improved the quality of the work. However, the article should still be treated as a communication (the authors confirmed that they are aware of the limitations of their work) and only published under this condition.
